# Reflections on the Biology of Cell Culture Models: Living on the Edge of Oxidative Metabolism in Cancer Cells

**DOI:** 10.3390/ijms24032717

**Published:** 2023-02-01

**Authors:** Alba Moran-Alvarez, Pedro Gonzalez-Menendez, Juan C. Mayo, Rosa M. Sainz

**Affiliations:** 1Departamento de Morfología y Biología Celular, School of Medicine, Julián Claveria 6, 33006 Oviedo, Spain; 2Instituto Universitario de Oncología del Principado de Asturias (IUOPA), University of Oviedo, 33006 Oviedo, Spain; 3Instituto de Investigación Sanitaria del Principado de Asturias (ISPA), Hospital Universitario Central de Asturias (HUCA), 33011 Oviedo, Spain

**Keywords:** metabolism, redox signaling, cell signaling, cell culture, 3D

## Abstract

Nowadays, the study of cell metabolism is a hot topic in cancer research. Many studies have used 2D conventional cell cultures for their simplicity and the facility to infer mechanisms. However, the limitations of bidimensional cell cultures to recreate architecture, mechanics, and cell communication between tumor cells and their environment, have forced the development of other more realistic in vitro methodologies. Therefore, the explosion of 3D culture techniques and the necessity to reduce animal experimentation to a minimum has attracted the attention of researchers in the field of cancer metabolism. Here, we revise the limitations of actual culture models and discuss the utility of several 3D culture techniques to resolve those limitations.

## 1. Introduction

Deregulation in cell metabolism has been highlighted as one of the hallmarks of cancer. Transformed cells increase their demand for nutrients, mainly glucose and glutamine, to support their high proliferation [1]. Since Otto Warburg described the metabolic shift that most tumours undergo, favoring aerobic glycolysis instead of the tricarboxylic acid cycle (TCA), deep characterization of tumour metabolism has been done in in vivo and in vitro systems [2,3].

Under physiological conditions, most cells oxidate carbon macromolecules. The electrons released by those reactions are carried by NADH and FADH_2_, which are recycled by the mitochondrial electron transport chain in aerobic respiration (OXPHOS). This process provides both adenosine triphosphate (ATP) and reducing power (NAD^+^ and FAD_2_^+^) [4]. However, tumour transformation increases the need for NAD^+^ and FAD_2_^+^, and carbon intermediates, while reducing oxygen and nutrient availability.

NADH and FADH_2_ levels surpass the recycling capability of mitochondrial OXPHOS, and together with hypoxia, reactive oxygen species (ROS) are increased. To deal with their new environment, neoplastic cells diversify their nutrient sources, for example, by taking other sugars than glucose, such as fructose or aminoacids, as glutamine; by recycling metabolic products of carbon oxidation, such as lactate, which is plentiful in the tumour milieu and often in distant regions; or by developing strategies of nutrient acquisition that are not utilized in healthy contexts [3].

The diversification in nutritional resources and products during cancer progression allowed the development of non-invasive imaging techniques for diagnosis and monitoring treatment response in research and clinic contexts [4]. Besides 1-[18F]fluoro-2-deoxy-D-glucose ([^18^F]FDG) uptakes in solid tumours, carbon-13- or nitrogen-15-labelled carbohydrates and amino acids have been employed in hyperpolarized nuclear magnetic resonance studies. Those have provided further opportunities to identify potential targets for new and adjuvant therapies that target cancer nutritional requirements [5,6,7,8,9,10].

Many of those metabolic characterizations have been assumed based on in vitro studies, which have proven to be suitable for demonstrating differences in metabolic demands and usage as well as modifications in the preferred metabolic routes. However, in vitro results often fail to mimic in vivo relevance and have even less clinical applicability [11].

Many differences between in vitro and in vivo results are due to the use of cell lines that do not reproduce the inner diversity of tumour cell populations and rapidly develop genetic and phenotypic alterations, inducing artefactual results [11,12,13]. An alternative to solve those problems is using primary cultures, but those must modify their phenotype to survive in the culture. They lose the proper context within the tumour microenvironment, which is not present in the culture plate. Only those cells that adapt to such altered conditions survive the process, which means an additional selection that reduces the similitudes between the culture and actual tumours [14].

Many of those differences between real tissue and 2D cell cultures are because cells impose oxidative stress to which they are forced to adapt. The primary source of oxidative stress is oxygen concentration (up to 18% in cultures and below 10% in inner tissues, being around 5% in most of them) [15]. Other ROS, such as H_2_O_2_, artificially increases in traditional culture plates. The amount of H_2_O_2_ varies regarding the utilized media, the pH, and the capacity of cells to produce molecules, such as oxalacetate, α-ketoglutarate, or pyruvate, that can act as antioxidants [16]. Not only is it difficult to ensure if metabolite production is then an artifact, but also cell sensitivity against antioxidant treatments such as ascorbate, thiols, tea, wine extracts, dopamine, or polyphenols has proven to be altered because of H_2_O_2_ and other antioxidants in media [11,17,18,19]. Hyperglycemia in several media also increases the generation of mitochondria superoxide (O_2_^−^), which further causes cells in the culture to adapt, increasing their inner antioxidant defenses. Some cell lines activate the redox state master regulator transcription factor NRF2 to confront those insults. They increase catalase production and stimulate cystine and cysteine membrane transport to scope oxidative stress [20,21,22].

Those misleading conclusions have raised the discussion about abandoning or not abandoning cell culture systems [23]. However, acknowledging these inaccuracies in the representativeness of cell culture runs concurrently with a low percentage of basic in vivo research translating into the clinics. Thus, the pressure to reduce the use of experimental animals has been increasing, motivating the development of alternative culture technologies that overcome the lack of representativity and allow us to characterize procedures that were only scrutinized in vivo [24,25].

Since early 2000, a plethora of 3D culture technologies has been developed. Either the first simple spheroids or the latest microfluidic devices to produce “cancer on a chip” offer an opportunity to overcome, at least in part, the weaknesses of 2D cultures in cancer metabolism studies. 3D structures create a gradient of O_2_ and nutrients and some mechanical tension absent in regular culture plates. When employing gels to culture over or inside them, some of the nutritional resources and mechanical stimuli that cells face in a real scenario can also be analyzed.

Since it is highly encouraged to transit into more faithful cultures that could reproduce human pathology in its metabolic and redox environment, we aim to explore which advantages 3D culture technologies offer over traditional 2D cell cultures, so they potentially become even a preferred model that allows the reduction of animal research. Here, we will revise those particular hallmarks of cancer metabolism that should or should not be explored using conventional 2D cell culture techniques and the proposed methods to avoid those limitations.

## 2. Differences in the Uptake of Nutrients between 2D and 3D Cell Cultures

Eukaryotic cells sustain their survival requirements mainly through glucose and glutamine uptake or their synthesis from other nutritional sources. Under physiological conditions, glucose uptake is a highly regulated process, one of whose main activators are growth factors. Thus, insulin, insulin-like growth factor (IGF1), or epidermal growth factor (EGF), among others, activate the tyrosine kinase receptor (RTK)/phosphatidylinositol-3-kinase (PI3K)/protein kinase B (AKT1) axis and cause glucose transporter 1 (GLUT1) translocation into the plasma membrane [4]. This pathway is commonly dysregulated in cancer because of genetic mutations or epigenetic alterations. Besides increasing glucose uptake through GLUT1, AKT1 phosphorylation also increases the activity of glycolytic enzymes such as hexokinase, which raises glycolysis, further enhancing carbon metabolism [26].

In the same way, the amino acid acquisition is also deregulated in cancer. Although glutamine is a non-essential amino acid under physiological circumstances, tumours become addicted [27]. Its uptake increases partially because of the higher production of its transporters ASCT2/SLC1A5 and SN2/SLC38A1. Their expression is upregulated by several oncogenes, such as MYC, RAS, and PI3Kα [28]. Once inside the cytosol, glutamine can be incorporated into the mitochondria through the SLC1A5 transporter, whose production is well-deregulated in tumours [29]. Glutamine is crucial for tumour progression since it fuels the synthesis of non-essential amino acids such as glutamate, α-ketoglutarate, and nucleotides [30].

Thus, nutrient scarcity constitutes a selective pressure that modulates cancer and tumour-associated cell behavior, which switch their preferred nutrient source and exhibit a plethora of nutrient acquisition mechanisms to sustain their growth. For instance, autophagy gets activated when nutrients are scarce, and cells degrade their organelles to transition and maintain survival [4]. Cancer cells can also emit plasma membrane projections to engulf extracellular macromolecules and vesicles as nutrient support. This process, called micropinocytosis, is employed to uptake proteins, collagen, and extracellular lipids [4,31]. Noticeably, serum albumin and fragments of type I and IV collagen are incorporated by this strategy in pancreatic cancer cells under low glucose or glutamine concentrations [32]. Another strategy to include extracellular materials is receptor-mediated endocytosis which allows cancer cells to incorporate cholesterol and transferrin, which is used to uptake iron [3,33]. Finally, transformed cells even use cell debris by a nutrient acquisition mechanism called entosis [34].

Another consequence of nutrient scarcity is the reduction in s-adenosine-L-methionine levels, which alters the activity of DNA and histone deacetylases, probing the connection between nutritional and epigenetic status and its role in satisfying cell growth demands [35,36].

Considering available knowledge about tumour cells’ nutritional demands, control of media composition, and their renewal, pH and oxidative status stand out as determinant factors in drawing conclusions based on in vitro studies. However, this is not only a matter of quantity; the interaction between tumour cells and other cell populations within their milieu determines their ability to achieve resources and progress into more aggressive phenotypes. For instance, amino acid depletion decreases T-cell activation in the tumour milieu since they require a proper supply of glutamine, glucose, serine, and alanine [37,38]. Other cells, such as adipocytes and nervous cells, also co-operate with tumour expansion; pancreatic neurons have been shown to provide amino acids for transformed pancreatic cells, while melanoma and prostate cancer cells use surrounding adipocytes as a source of fatty acids [39,40,41].

Therefore, how cancer cells conduct nutrient uptake is a multifactorial process involving the microenvironment and unique cell adaptation strategies. Here, experimental procedures should be performed carefully since it is difficult to reproduce the delicate interactions present in vivo. In fact, most culture media compositions alter those processes in cultured cells.

As expected, metabolic studies have revealed differences in glucose and amino acid uptake when comparing 2D and 3D culturing techniques. However, metabolomic profiles are more similar between 3D cultures and xenografts than between those and 2D traditional cell cultures. For example, 3D cultures of prostate and bladder cancer cell lines have increased upstream glycolytic and TCA metabolites, whereas continuous 2D cultures are mainly glycolytic [42]. Those experiments have also revealed a higher production of amino acids in 3D structures, mainly glutamate and glutamine, that are employed to synthetize other non-essential amino acids, glutathione, and nucleotides. Amino acid production by 3D systems also differs from non-tumoral cells [42]. 3D constructions of colorectal, lung, breast, and ovary cell lines further support the increase in amino acid acquisition and biosynthesis, and upregulation in its interconnected TCA pathways, urea cycle, and pyruvate biosynthesis. In addition, organotypic cultures of those cell lines reveal a higher plasticity than their 2D counterparts concerning glutamine addiction [43] (Table 1).

The consequences of nutrient uptake go beyond tumour growth rate; they also affect their capacity to overcome insults. This has been demonstrated in experiments with head and neck cancer cells, where differences in glutamine metabolism after chemotherapy have been found when cultured in regular plates vs. scaffolds. In 3D cultures, only the conversion of glutamine in glutamate seems critical in the cell response to the treatment with mitochondrial metabolism inhibitors [52].

The increase in amino acid acquisition and biosynthesis in 3D cultures vs. 2D also makes them a more suitable model to analyze mTORC1-driven pathways in cancer, which has proven to be a bona fide drug target in many tumours [53]. Regular 2D cultures reduce the number of building blocks available and cause organellar oxidative damage due to over-physiological oxygen exposure. Those basal stressing conditions reduce cancer cell capability to show their resilience in vivo, in which the mTORC1 pathway partakes. Successful research has been done incorporating 3D cultures. They have tested AKT inhibitors in 3D cultures of colorectal cell lines, and the synergy effects were found to reproduce in patient-derived xenograft models [60]. Targeting mTORC1 has been a successful strategy to avoid cell survival and migration in the 3D ovary and gastric cancer cultures, further proving this culture technique’s utility to target tumour metabolic shifts [54,65] (Table 1).

Glucose, glutamine, and oxygen scarcity are also strong autophagy drivers. In addition, three-dimensional tumour cell growth stimulates autophagy and reproduces resistance against chemotherapy in neuroblastoma cell lines [66]. Even more, 3D experiments have revealed that dormant breast cancer cells employ autophagy to survive redox stress conditions and promote metastasis, a phenomenon reproduced in vivo [56].

Other nutrient acquisition strategies also differ in 3D and 2D cultures. Thus, it has been shown that micropinocytosis only occurs in specific areas of 3D culture structures of HeLa cells. In contrast, every cell can use this strategy when the culture is monolayer [58]. In addition, prostate cancer cells’ 3D cultures exhibit higher resistance than their 2D counterparts [67]. 3D cultures of breast cancer and in prostate, pancreas, lung, colorectal, and bladder tumours also report the use of micropinocytosis to resist doxorubicin, gemcitabine, and 5-fluorouracil by incorporating extracellular proteins or amino acids to overcome stress induction [68].

Those results demonstrate the relevance of cell-to-cell and cell-to-matrix interactions regarding nutrient uptake. Importantly, tumour cells can better face nutritional stress in 3D complex structures, at least partially because they are not forced to favor the glycolytic pathway as they do in traditional 2D cultures, so they can use their metabolic flexibility to improve their resilience. Instead, they change their main nutritional sources or implement other strategies such as autophagy, entosis, or macropinocytosis. Therefore, 3D constructs are much better representatives of the adaptive potential of cancer cells that drug testing and metabolic experiments should no longer ignore.

## 3. Extracellular Matrix-Embedded Cultures Allow Us to Better Understand Cell Plasticity in Cancer

Extracellular matrix remodeling affects cell metabolic pathways. The uniqueness of the tumour microenvironment is also highly relevant for reproducing the metabolic adaptations that cancer cells undergo in vivo. Cells inside tissues are also subject to mechanical pressures that determine their success in colonizing metastatic locations. For instance, the employment of the extracellular matrix in the culture and the development of devices that recreate mechanical stimuli cancer cells face in vivo have helped to analyze the mechanism melanoma cells use to colonize lung tissue [65,69]. Regarding metabolism, it has been demonstrated that extracellular hyaluronic acid increases GLUT1 production and glucose acquisition and upregulates glycolysis [38,39]. Hyaluronic acid degradation causes a reduction in thioredoxin-interacting protein (TXNIP) levels in cancer cells. When enough nutrients are available, TXNIP retains GLUT1 in cytosolic vesicles. At the same time—while when they are scarce, it is degraded—GLUT1 moves to the cell membrane, and glucose uptake rises, concomitantly with an invasive phenotype acquisition [56]. Different extracellular matrix compositions are also highly relevant in the acquiring of migration phenotypes. This variability might be partially due to the further collagen production by tumour-associated fibroblasts. Those differences in collagen deposition promote invasiveness in breast and pancreatic cells [64,65]. Consequently, extracellular matrix composition should be generated to analyze metastatic potential properly. Primary culture of patient-derived organoids is the best option to develop such a representative environment, as found in those constructs [44,59].

It is then crucial to incorporate cell-to-cell and cell-to-matrix interactions when evaluating these opportunistic nutrient-scavenging strategies. Differences in nutrient and oxygen availability determine how cancer cells take advantage of the disposable resources in their milieu, where the extracellular matrix actively partakes. This is even more relevant when evaluating treatment efficacy in vitro, which could differ depending on the use of those strategies by transformed cells.

## 4. Adaptations in Carbon and Amino Acid Metabolism to Support Biosynthesis

One of the significant metabolic transformations in cancers is the shift towards glycolysis in the presence of oxygen, named the Warburg effect, which provides cancer cells several advantages. First, they produce ATP faster and low pH in the microenvironment, which facilitates proliferation, increases NAD^+^, and provides building blocks for anabolism supporting cell growth [70]. Then, TCA intermediates are used as carbon substrates for biosynthesis, driven by the PI3K-AKT-mTORC1 pathway [60]. When this metabolic transformation occurs, citrate and glutamate get out of the mitochondria and are used to fuel lipogenesis and to synthesize α-ketoglutarate. An increase in α-ketoglutarate levels results in mTORC1 activation, which blocks autophagy and further promotes aerobic glycolysis and protein translation [28,71].

The most apparent consequence of the Warburg effect is the increase in lactate production, which is employed as a carbon source for biosynthesis in neighbouring cells. Lactate, produced from the tumour, microenvironment components, or distant cells, is another carbon metabolite that supports biosynthesis. Circulating lactate has been shown to support TCA, as demonstrated by ^13^C labelled experiments in lung, brain, and renal cancers [72,73,74]. In addition, the conversion of pyruvate into lactate regenerates NAD^+^ while its secretion acidifies the tumour milieu, promoting proliferation [75].

Not only cancer cells undergo the Warburg effect. For instance, it has been demonstrated that colorectal cancer cells maintain their energy supply by inducing the Warburg effect in cancer-associated fibroblasts (CAFs), which secrete lactate and pyruvate that is further employed by tumour cells that maintain OXPHOS and TCA in the so-called reverse Warburg effect [76]. In addition, the immune compartment gets affected by those metabolic alterations. The increased lactate production and acidification also promote the transformation of myeloid cells, which become pro-cancerous instead of antitumoral. For instance, transformed macrophages secrete proteases that reduce the amino acidic component of the extracellular matrix, which becomes immunosuppressive [3,77].

Traditional culture conditions increase glycolysis on their own, even in non-transformed cells [78]. However, when nutritional and gas gradients present in 3D cultures are incorporated, more representative results emerge. For example, the capability of tumour cells to favour glycolysis and reuse TCA intermediates for biosynthesis under mitochondrial energy stress is higher in organotypic cultures of colon, lung, breast, and ovary cells than in 2D cultures. In addition, they show higher levels of amino acids, increase lipids and pyruvate, and enhance urea cycle activity. Those changes make organotypic cultures less dependent on glutamine acquisition [43]. This is relevant because, in a real scenario, not all cells are supplied with the same glutamine concentration present in culture media.

Therefore, tumour cells modify the metabolic flux redirecting carbon intermediates according to their availability and the environmental stressors they are exposed to. In natural tissues, glycolysis upregulation is a useful tool, either in cancer cells or in the transformed microenvironment. However, traditional 2D cultures lack the proper stimuli to induce that switch, partly because of the high basal oxygen concentration and the absence of a real 3D complex environment. 3D cultures and oxygen flux control are suitable for partially reproducing that environment and evaluating potential metabolic therapeutic changes (Figure 1).

## 5. Metabolic Adaptations Help to Face Increased Oxidative Stress in Cancer Cells

Oxygen has been one of the main drivers of evolution under both physiological and pathological scenarios. Cell survival depends on energy supply, mainly sustained by ATP synthesis through OXPHOS. Thus, changes in oxygen and oxygen reactive species (ROS) levels are critical in developing transformed phenotypes.

It has been reported that cancer cell growth leads to an increase in ROS and, consequently, damages cell macromolecules. That is particularly true in the core of solid tumours where the supply of oxygen and nutrients are reduced. Therefore, a decline in the NAD^+^/NADH, FAD_2_^+^/FADH_2_, and NADP^+^/NADPH ratios is found. This is partially due to the increased demand for reduction power to eliminate the electrons liberated during ATP synthesis in mitochondrial. When ATP demands rise, as occurs during cell growth, mitochondria increase their efforts to produce it, but cell metabolism must adapt when ROS production gets too high [3,4]. Thus, Warburg metabolism, favouring glycolysis to generate ATP, reduces mitochondrial activity, decreasing ROS production.

However, genetic alterations favour the capability of cells to face redox stress during cancer progression. During hypoxia, HIF1α, NRF2, and KEAP1 are activated, which has aberrant consequences for cancer cells. They promote aggressiveness and resistance to cell death [79]. Many tumours, because of their activation, enhance the production of antioxidant enzymes such as thioredoxin (TRX), superoxide dismutase (SOD), or catalase [80].

The reduced entry of pyruvate into TCA and the secretion of lactate increase when oxygen concentration is low, also due to the activation of HIF1α and NRF2, not only in tumours but also in non-tumour associated cells [81,82,83]. Under hypoxia, pyruvate can be converted into L-2-hydroxyglutarate instead of lactate. The accumulation of this oncometabolite results in HIF1α stabilization, [84]. In addition, within the hypoxic core of solid tumours, HIF2α stabilization drives the conversion of α-ketoglutarate into citrate to support fatty acid synthesis, regenerating NAD^+^ and FAD_2_^+^ [85,86,87]. Oxaloacetate is transformed into malate that gets out of mitochondria and is used in pyruvate synthesis, generating NADP^+^ [75]. The activation of NRF2 driven by hypoxia is also implicated in regulating nutrient flux inside the cell. It has been demonstrated that glutamine is one of the main sources of carbon for glutathione synthesis since it was found enriched when cells were supplied with ^13^C-labelled glutamine [28,29,88]. Glutathione synthesis from glutamine, whose uptake is increased under hypoxia, is enhanced in cancer cells.

The interaction between redox sensing proteins and epigenetics is another modulated process in cancer. Epigenetic repression of the HIF inhibiting factor (FIH) should be highlighted [89]. For example, low glutamine availability in the hypoxic core of solid tumours leads to a decrease in α-ketoglutarate synthesis, which reduces histone demethylases, resulting in dedifferentiation due to epigenetic alterations [90]. Then, redox signalling also regulates nutrient availability and epigenetic mechanisms that govern cell fate.

The differentiation status of tumour cells is also controlled by oxygen concentration and its consequences in epigenetics [90]. For instance, DNA demethylation ten-eleven translocation (TET) enzymes are regulated by oxygen levels, which is crucial in maintaining pluripotency in both physiological and carcinogenic scenarios [91]. It has been demonstrated that the activity of those enzymes is reduced under oxygen concentrations below 5%. The apparition of hypermethylation clusters in breast, kidney, and cancer is associated with tumour dedifferentiation and malignization [92,93]. Furthermore, enzymatic demethylation Fe(II)- and 2-oxoglutarate-dependent dioxygenases are highly dependent on iron availability as a cofactor and molecular oxygen. Oxygen concentration regulates iron oxidation status, which is tissue-dependent and highly variable during tumour development [35,94]. Ascorbate is a key cofactor of demethylation Fe(II)-and 2-oxoglutarate-dependent dioxygenases. It contributes to maintaining an adequate redox environment, so there is enough iron. When the oxygen concentration is over-physiological, ascorbate oxidises, even causing cell death in the culture, but this scenario would never be found in vivo. High O_2_ levels also increase transferrin synthesis, so iron uptake surpasses in vivo acquisition ratio [11,33,95]. Nitrogen reactive species, such as ROS, are also increased in that context, altering the activity of MAPK, and the PI3K-AKT pathway, and the activity of relevant transcription factors such as NRF2, NFKB, and STAT3 [96].

In vivo cell detachment from the extracellular matrix also increases ROS levels that are counteracted through different mechanisms. For example, many cancer cells form clusters with a hypoxic core where glycolysis is favoured, and autophagy activated to recycle damaged organelles [97]. Other studies suggest the migration phenotype could also be a survival strategy against redox stress in which tumour cells promote glucose acquisition by increasing GLUT1 levels and relocation [98,99,100].

Tissue vascularization is also determinant in cancer progression and metastasis. Oxygen and nitric oxide regulate neovascularization in the core of solid tumours. Hypoxia promotes the secretion of vascular endothelial growth factor (VEGF) by tumour cells and by tumour-associated macrophages [77]. Furthermore, the hypoxic environment also helps to create an immunosuppressive environment in the tumour microenvironment when scarce oxygen causes the activity of T-cells to decay because their mitochondrial activity becomes compromised [101].

## 6. Differences in Metabolic Adaptations to Face Redox Stress between 3D and Traditional Cultures

Despite the increasing number of papers that report information about redox control in cancer, experts in redox studies express strong critiques of most of the techniques employed in cell culture experiments. As it has been elegantly noticed by K. Mootha, Giovanni E. Mann, and B. Halliwell, among others, some aspects of basic cell biology are obscured when oxygen concentrations are altered, as happens in the commonly employed 21% oxygen concentration in cell culture experiments. Inside cells plates, the oxidative environment forces cells to adapt, making those transitional responses in specific regions in vivo become the standard scenario in vitro (Table 1) [21,78]

Then, when studying how cancer cells cover their necessities of electron acceptors, experiments should ensure physiological oxygen concentrations, even the hypoxic condition that drives HIF1α and NRF2 activity in the core of solid tumours. Traditional 2D cultures show significant weaknesses in this regard. A higher mutation rate is registered in the cell culture, partially because of the DNA oxidative damage. Hyperoxia controls metabolic flux because many enzymes use oxygen as a substrate [95]. HIF1α, NRF2, and other redox control regulators are naturally activated under hypoxia and control tumour progression, aggressiveness, and its response to treatment. However, traditional culture conditions completely mask their real activity [21,98,99,102].

When the oxygen concentration is too high, it can also react with metals, such as iron, damaging essential cofactors through the Fenton reaction and promoting cell death. In contrast, in vivo, those same cells could progress to metastasis. High O_2_ levels artificially increase transferrin production in cancer cells to survive [11,33,95].

Not only does the oxygen concentration determine cell redox transformations in vitro, but culture media also increases the variety of alterations observed. They are often deficient in antioxidants, and many of the antioxidants supplied, for instance, polyphenolic compounds such as ascorbate and flavonoids, are unstable and convert into oxidants. This has led to misleading conclusions about the potential of some nutraceuticals to cause cancer cell death [11,17]. On the other hand, cells dramatically enhance antioxidant production and secretion, such as glutathione, α-ketoglutarate, and oxalacetate, since they readily scavenge the supraphysiological H_2_O_2_ levels [11,78].

Glucose and glutamine concentrations in vitro also exert physiological ones, which increase mitochondrial ROS production and can cause post-traditional protein alterations which affect their function [11].

Exacerbated oxidative stress has proven to cause telomere shortening, which also contributes to carcinogenesis, through an increased mutational rate [103]. Comparative studies in HeLa cells have revealed high variability between the same cell line upon passages and among different labs. This results in genetic instability and clonal selection, producing a variation in copy number and causing gene expression variability [13]. Similar alterations have been found in MCF-7 and HEK293T cell lines [104,105]. This transformation occurs more slowly through cell passages when cells are maintained under a low oxygen concentration [95].

The fine-tuning of metabolic regulation by redox and nutrients is challenging to model in vitro. As previously exposed, hyperoxia and media composition disrupt the normal functioning of redox defences in cancer cells that become glycolytic and lack the flexibility to adapt as they show in vivo. The use of hypoxic chambers has undoubtedly been beneficial in revealing the relevance of oxygen levels when making conclusions about tumour metabolism. However, more accuracy is required to have translational results since the nutrient concentration, media composition, and cell microenvironment are also determinants. 3D culture has proven to be a useful tool to understand how cancer cells respond to redox stress since they are exposed to an oxygen gradient with a hypoxic core where HIF1α activates. In this regard, HIF1α drives metabolic changes that allow cancer cells to produce more ATP while maintaining a low NAD(P)H/NAD(P)^+^ ratio required to ensure mitochondrial activity. Such metabolic differences have also been reported in 3D cultures of ovarian, colon, breast, pancreatic, lung, and liver cancer. The connection between the preference for assigning glucose to TCA instead of glycolysis has been related to differential sensitivity to treatments since it is also associated with cell capability to maintain redox balance [44,45,46,47,48,49,51,52,106].

It is worth noting that the experimental strategies to measure ROS and antioxidants should be improved. 3D culture, particularly organotypic cultures, has proven to be helpful for studying cancer metabolic adaptation to hypoxia and its connection to treatment response because they reproduce some of the main gradients that drive malignization.

## 7. 3D Models to Improve Cancer Cell Metabolism Research

In cancer metabolism, many experiments are still conducted in 2D culture, xenografts, or mice models. Traditional culture conditions cause the loss of cell polarization and change their response to stimuli. On the other hand, 3D culture retains most stimuli, including the interphase between tissues and the spatial and temporal gradient of nutrients and oxygen (Figure 2). Thus, 3D constructions could bridge the gap between oversimplified 2D cultures and under-representative animal models to improve our understanding of cancer cell metabolism [107,108,109].

Most 3D culture techniques are either challenging to reproduce, as it occurs with patient-derived organoids, or can only model a limited number of aspects in vivo. For example, spheroid cultures often lack vasculature or nerve signalling. However, during the last decade, complex culture strategies have been developed to function as a helpful tool for drug discovery and testing, and to improve personalized medicine [110].

3D culture systems could be grouped into three categories: multicellular tumour spheroids, patient tissue-derived organoids, and tumour on a chip, each with different pros and cons regarding experimental reproducibility and tumour metabolism representativeness.

Tumour multicellular spheroids can be constituted by tumour cells or a combination of different cell types in the microenvironment. They can exhibit different morphologies, from round to stellate depending on culture method and cell characteristics (Figure 2A) [111]. One of the most straightforward but laborious strategies is hanging drops. In this technique, cell suspension drops that could be embedded in an extracellular matrix, such as Matrigel, are seeded over the top of a traditional culture dish, on a low-adherence culture plate, or in scaffolds designed for that purpose. Gravity makes cells aggregate, creating a rounded spheroid in which a gradient of oxygen, nutrients, and metabolic waste products are produced. The centre of the structure has reduced availability of both nutrients and oxygen, whereas it accumulates most of the metabolic residues. Those structures also exhibit a different rate of cell survival and proliferation. The outer ring is the most proliferative. In the middle layer, quiescent cells are found; meanwhile, in the core of spheroids, hypoxia is developed, causing necrosis and metabolic adaptations in the surrounding concentric cell coats [110]. Spheroids can also be generated by seeding cells over or inside hydrogels made of natural or synthetic polymers. Those strategies partially reproduce cell-to-matrix interactions since the mechanical stimuli, or the nutritional composition of the extracellular matrix could be recreated. Hydrogel properties can be adjusted by controlling pH, temperature, light exposure, or electric and magnetic stimuli [111]. They can also be created through a non-attached culture in a rotating or spinner flask, thus exploring cancer cell capacity to survive in circulating clusters, similarly to what they find in the systemic circulation during metastasis [112]. Several metabolic changes occur since tumour cells have to profoundly transform their metabolism to avoid anoikis caused due to matrix detachment to solve oxidative stress, for example, by upregulating autophagy [56,57] (Table 1).

Spheroids have been the most employed 3D culture strategy besides patient-derived organoids. In bigger spheroids (500 μm in diameter), the microenvironment of micro-metastasis and vascular tumours is reproduced. Therefore, they help understand the role of hypoxia in tumour transformation and development [110,113]. The hypoxic core of spheroids shows gene expression changes that control nutrient uptake, similar to in vitro solid tumors. The activation of AMPK, the reduction of TXNIP levels, and the activation of cell response to hypoxia through the HIF1α or NRF2 pathway led to an increase in GLUT1 levels and glucose uptake. Those alterations that imply metabolic rewiring during low oxygen concentration in solid tumours’ cores have been found in spheroids of the bladder, prostate, ovary, breast, pancreatic, lung, and liver [42,45,46,47,48,50,51].

Multicellular spheroids have also allowed an understanding of the role of vascularization in tumour progression mediated by CAFs. Furthermore, they have been employed as a model to understand the role of extracellular matrix stiffness on vascularization induction in stromal breast cancer [114]. Regarding the immune compartment, spheroid generation of non-small cell lung carcinomas, including CAFs and monocytes, have been generated through microencapsulation. Producing cytokines such as IL5, IL10, CCL22, CCL24, or CXCL1 generates an immunosuppressive atmosphere inside the culture [115].

Along with a wide variety of advantages in terms of representativeness of cell-to-cell and cell-to-matrix interaction, 3D cultures offer some additional challenges that should be carefully addressed. As we have previously highlighted, one of their major advantages is the chance to analyse metabolic adaptations to hypoxia within a gas and metabolic gradient. While in 2D cultures gas exchange vary depending on the amount of medium employed, the frequency of its replacement, and on cellular rate of consumption, in 3D constructs cell oxygenation can be controlled with different approaches. Oxygenation can vary depending on hydrogel or polymer gas permeability in embedded 3D structures [116]. Despite the relevance of this topic, little attention is often paid to provide pericellular oxygen concentration and most articles only report data from incubators’ setup. Several methods can be employed to get more representative measurements such as oxygen electrodes that can be already placed in the culture plate and, in 3D hydrogels, fluorescence quenching probes that indirectly indicate the amount of O_2_ and can provide a 3D oxygen concentration map through the construct [117,118].

Not only can oxygen be retained in polymers and hydrogels, but much of nutrient availability also gets affected by its composition. For instance, polydimethylsiloxane (PDMS), which is commonly employed in microfluidic devices, due to its low price and flexibility which allow us to reproduce mechanical stimuli that exist in vivo, also absorbs small hydrophobic molecules as a counterpart [119]. Huge efforts have been made to generate polymers and hydrogels that mimic physiological nutrient flux, such as the development of tetrafluoroethylene−propylene, that offers the same properties that PDMS does without retaining hydrophobic probes, or collagen vitrigel that allows better protein interchange between cell populations that can be cultured on both sides [119,120].

Permeability is only one of the multiple physical and chemical parameters that are considered in the design and selection of devices and materials in 3D culture. Since cancer cell response to treatment can be conditioned by mechanical stimuli in cell environment, and drugs can also modulate the cell’s mechanical properties on their own, the interest in intersectional studies in rheology and cell biology has increased [120,121,122,123,124]. Therefore, the microarchitecture of hydrogels rises as a major field of study and innovation since their elasticity (how they deform/recover their shape when a force is applied, i.e., their elastic and shear modulus), their porosity (the shape, size, and distribution of those pores), their composition (dimensions and disposition of fibres) and viscosity interfere in cell shape, migration, and gene expression [125]. Noticeably, HIF1a stabilization can increase due to drug-induced EMC stiffness in breast cancer, which might implicate that mechanical stimuli should be considered when understanding the genetic response in 3D cancer cultures [126].

To characterize hydrogel architecture, several electron- and photon-based technologies have been employed. For example, scanning electron microscopy, probably the most used, is suitable to explore hydrogel porosity and allow us to also incorporate X-ray-based tools that provide information about cultured cells. However, processing samples often require dehydration or vitrification, so live cells cannot be monitored [127]. On the other hand, environmental scanning electron microscopy allows us to employ hydrated samples due to the use of vacuum, but it has lower resolution and condensation, and exposition to voltage may cause artefacts [125,128]. Some photon-based devices provide a strategy to understand hydrogel containing living cells, such as confocal laser microscopy, that yields high-resolution 2D and 3D images either employing fluorescence probes or auto-fluorescence hydrogels; or second harmonic generation, where signals are produced by macromolecules after an incidence with non-destructive laser and avoiding the use of fluorophores [125,129]. Importantly, traction force microscopy and atomic force microscopy offer the chance not only to measure hydrogel topology, but also to characterise its physical properties such as elasticity and response to small forces such as those that occur within real tissue in 2D and 3D constructs [130,131,132].

Altogether, the interaction between cancer cells and the actual device in which they are cultured rises as a new significant factor to be consider in in vitro research.

## 8. Conclusions

Despite the utility of the spheroid structure, more complex culture structures are required to understand cancer cells’ metabolic transformation further. Patient-derived organoids reproduce cell diversity and the tumour microenvironment structure (Figure 2B). However, they have been mainly employed to improve drug testing responses in colon and breast cancer. They have also helped identify cell populations relevant to co-operate in tumour progression, such as a certain fibroblast that produces IL6 in pancreatic cancer [133]. They constitute one of the most faithful reproductions of in vivo scenarios regarding cancer metabolism research. For example, intestinal organoids have demonstrated how cancer cells use their neighbouring healthy cells as nutritional resources. Reproducibility is one of the main challenges for these techniques. Organoids can be developed from embryonic cells, pluripotent stem cells, and adult stem cell. The last of them are the most useful in cancer metabolism study. They can be generated through different strategies, by disgregating adult tissue and allowing cells to group and reconstitute structures that reproduce the in vivo epithelial architecture; incorporating an air/liquid interphase in Boyden chambers where cells get nutrients through a semipermeable membrane, allowing us to study stromal and epithelial cells at the same time; or creating tissue-embedded droplets in spinning bioreactors. All those techniques require the inclusion of a growth factors cocktail that is tissue specific and usually include Wnt activators, receptor tyrosine kinase ligands, bone morphogenic protein inhibitor noggin, and TGF-b inhibitor [134]. In addition, imaging techniques such as microscopy and flow cytometry might be challenging to perform in some 3D cultures; they can require employing expensive devices or destroying the extracellular structure before using fluorescent probes to understand the metabolic and redox landscape they develop [59,110,134].

The use of 3D scaffold structures has set the bases to develop new culture strategies, such as 3D bioprinting, that offer a well-defined cytoarchitecture and composition. The techniques commonly employed to create those structures include extrusion, inkjet, stereolithography, laser, and electrospinning-based bioprinting [135]. Those structures will also be helpful in better understanding the relationships between different cell populations (Figure 3A). For instance, they allow us to investigate the transformation of CAFs by induction of reverse Warburg. Relevant discoveries were made in 3D bioprinting constructs of breast cancer cells cocultured with endothelial cells and CAFs, in which breast cancer cells cause the transformation of fibroblasts. In addition, in 3D scaffolds, it has been shown how pancreatic cells secrete TGFβ transform CAFs to support their nutritional demands. They also allow the control of other variables, such as time, that are highly relevant in tumour cell adaptations to nutrient and oxygen scarcity [136].

The recent incorporation of microfluidics partially reproduces the pathophysiological scenario in vivo, recreating the chemical gradients and mechanical stimuli. Although the development of plastic structures that mimic tumour, stroma, and vasculature might be technically complex and, depending on the material employed, also expensive, they have successfully represented how breast cancer promotes immune suppression and recreate oxygenation gradient inducing VEGF secretion. They have even reproduced cell migration strategies through the vasculature in lung cancer [137,138,139]. Therefore, those devices acutely show the variety of nutrient acquisition strategies mentioned above (Figure 3B).

Therefore, 3D culture has become an irreplaceable tool to improve cancer metabolism research in vitro and its consequences in cell response to therapy, allowing us to reduce the amount of animal studies and enabling translational research, and facilitating personalised medicine.

Recent advances in cancer research that have been exposed in recent years show the relevance of the redox environment in cancer development, progression, and metastasis formation. Tumour cells exhibit many metabolic adaptations involving genetic, enzymatic, and cellular responses to adapt to nutritional, mechanical, and redox insults they find in vivo [1,3,100]. Those alterations are difficult to reproduce in the traditional 2D culture approaches. It is worth noting that the over-physiological oxygen concentrations cell cultures are exposed to have received severe criticism from the experts in the field of redox biology, who have pointed out that the results obtained in many toxicological studies are affected by culture conditions [16,17,21,23,78,95]. The hyperoxia that cells are exposed to in traditional cultures forces them to acquire a glycolytic profile. In vivo, it is only one among many metabolic strategies they can employ to face redox stress. Therefore, we employ the utility of 3D culture techniques where oxygen concentrations, at least inside the structure, are more like the natural state. They have sought to reproduce the metabolic reprogramming of cancer cells and their microenvironment to sustain tumour progression. In this review, we conclude that those strategies are a useful tool to understand how those metabolic shifts allow cancer cells to develop resistance against therapy in a more realistic scenario, without the unnecessary employ of animals or the questioned use of conventional 2D culture strategies [103,104,139].

## Figures and Tables

**Figure 1 ijms-24-02717-f001:**
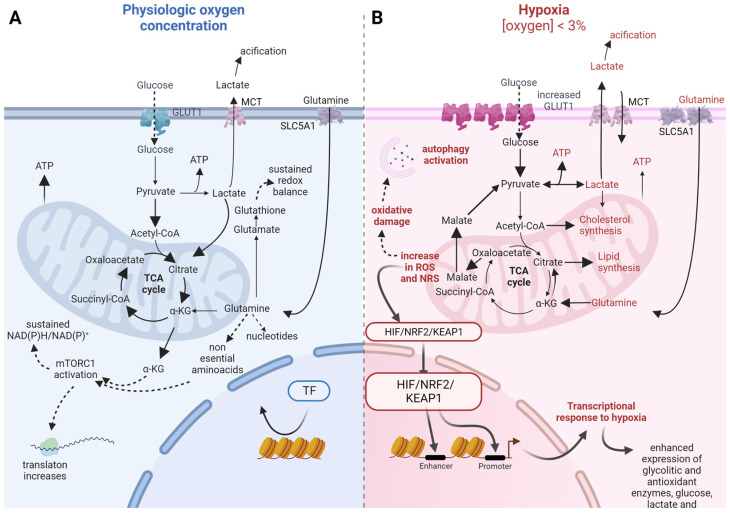
Metabolic adaptations to hypoxia in 3D cultures. Differences in nutrient uptake, main metabolic pathways, biosynthesis, and redox balance between physiological oxygen concentration (**A**, left blue scheme) and under hypoxia (**B**, right pink scheme). Solid arrows indicate nutrient flux; if wide, they show increased flux towards those pathways. Dashed arrows indicate indirect causal relationship. Faded color arrows indicate direct causal relationship between processes. GLUT1, glucose transporter 1; MCT, monocarboxylate transporter complex; SLC5A1, solute carrier family 5 member 1; TF, transcription Factor; TCA, tricarboxylic acid cycle; KG, alpha-ketoglutarate; mTORC1, mammalian target of rapamycin complex 1; HIF, hypoxia inducible factor; NRF2, nuclear factor (Erythroid-derived 2)-like 2; KEAP1, Kelch-like ECH-associated protein 1.

**Figure 2 ijms-24-02717-f002:**
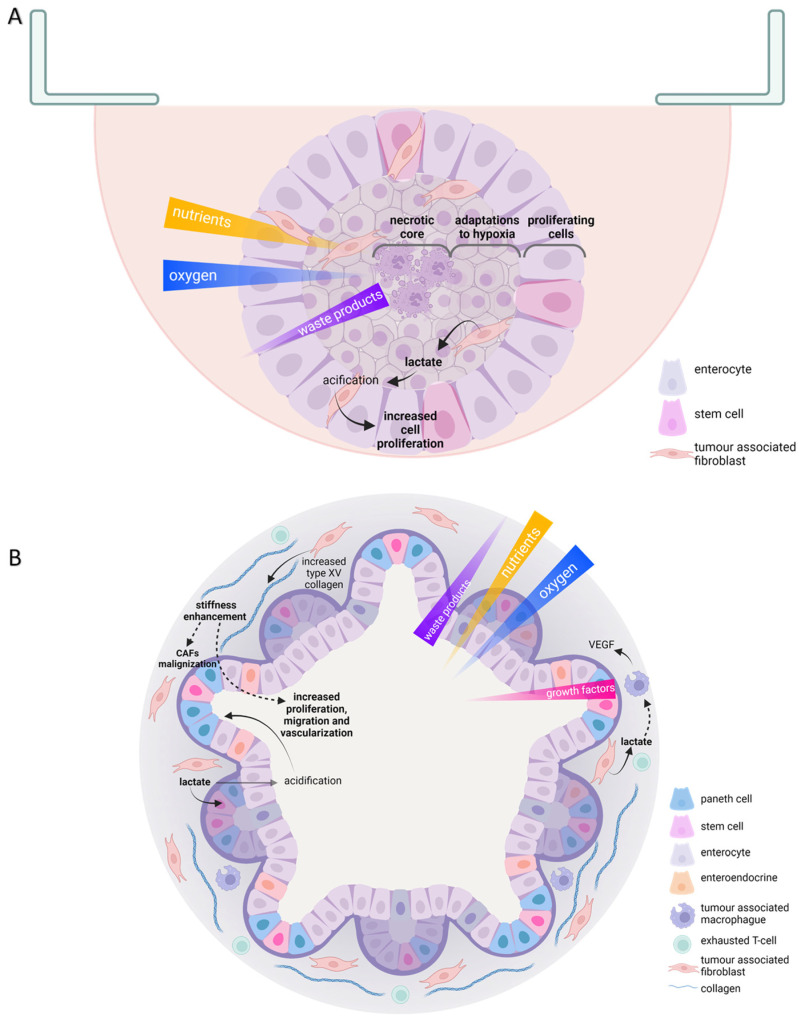
Useful scaffold-free 3D culture features in tumor metabolism study. Schematic representations of hanging drop multicellular intestinal spheroid (**A**); patient-derived colorectal organoid (**B**). Oxygen, nutrient, waste products, and growth factors gradients are indicated; color opacity and triangle width indicate high concentration. Cell types are indicated within each scheme legend. Solid arrows represent direct effects; discontinued arrows indicate indirect effects. CAFs, cancer associated fibroblasts; VEGF, vascular endothelial growth factor.

**Figure 3 ijms-24-02717-f003:**
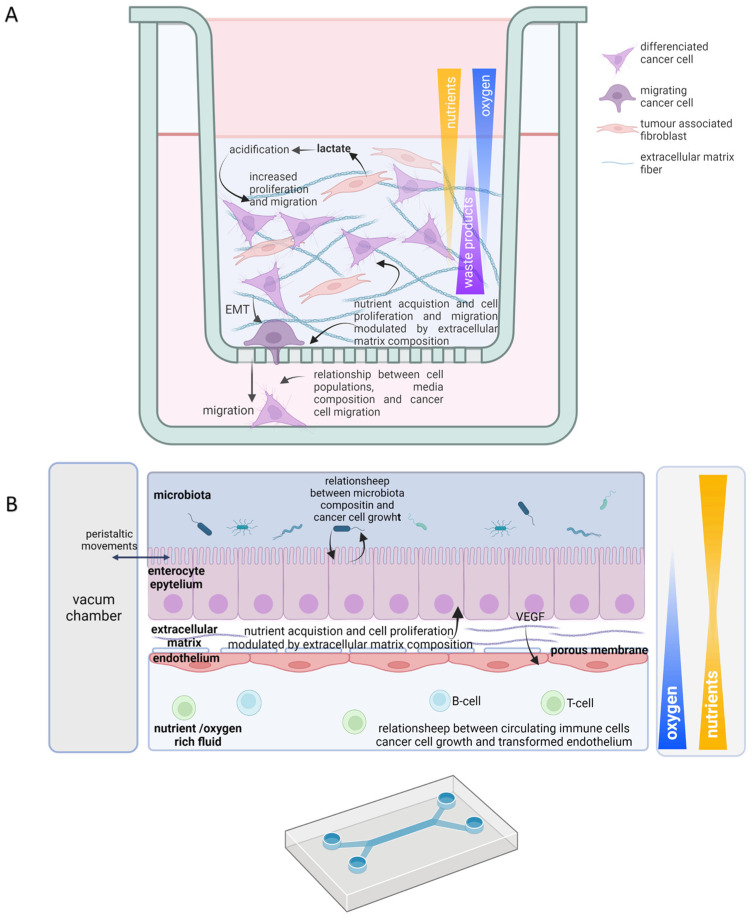
Useful scaffold 3D culture features in tumor metabolism study. Schematic representations of 3D-matrix-embedded scaffold co-culture of Hela cells and fibroblasts (**A**) and “gut on a chip” culture in a microfluidic device (**B**). Oxygen, nutrient, waste products, and growth factors gradients are indicated; color opacity and triangle width indicate high concentration. Cell types are indicated within each scheme legend. Solid arrows represent direct effects; discontinued arrows indicate indirect effects. EMT, epithelial mesenchymal transition; VEGF, vascular endothelial growth factor.

**Table 1 ijms-24-02717-t001:** Experimental discrepant result between 2D and 3D cultures.

Discrepancy between 3D vs. 2D Culture	3D Culture Technique	Cancer Type	Related Hallmark of Cancer Metabolism	Refs.
**Increased glycolysis upstream metabolites and TCA**	3D silica scaffolds (CellBed^®^)	Prostate Bladder	Deregulated uptake of glucose and amino acids Use of carbon metabolism to support biosynthesis	[42]
**HIF1a stabilization increases ATP production and sustains NAD(P)H/NAD(P)^+^ ratio**	Organoids	Ovarian	Deregulated uptake of glucose and amino acids Increased demand of electron acceptors Elevated reliance on oxidative stress protection mechanisms	[44]
	Spheroids	Ovarian Breast Colon Liver		[45,46,47,48]
	3D bioprinting	Breast Colon		[49,50]
	Magnetic 3D bioprinted spheroids	Pancreatic Lung		[51]
**Increased amino acid production**	3D silica scaffolds (CellBed^®^)	Prostate Bladder	Deregulated uptake of glucose and amino acids Use of carbon metabolism to support biosynthesis Increased demand for nitrogen Increased demand of electron acceptors Elevated reliance on oxidative stress protection mechanisms	[42]
**Increased glutamate synthesis from glutamine**	3D scaffolds (SeedEZ™)	Head and neck	Increased demand for nitrogen Increased demand of electron acceptors Elevated reliance on oxidative stress protection mechanisms	[52]
**Increased amino acid acquisition and biosynthesis**	Organotypic cultures	Colorectal Lung Breast Ovarian	Deregulated uptake of glucose and amino acids Use of carbon metabolism to support biosynthesis Increased demand for nitrogen	[43]
**Increased interconnexion between TCA, urea cycle, amino acids, and pyruvate synthesis**	Organotypic cultures	Colorectal Lung Breast Ovarian	Use of carbon metabolism to support biosynthesis Increased demand for nitrogen	[43]
**Reduced dependence on glutamine acquisition**	Organotypic cultures	Colorectal Lung Breast Ovarian	Deregulated uptake of glucose and amino acids Use of carbon metabolism to support biosynthesis Increased demand for nitrogen	[43]
**Increased dependence on mTORC1 activation**	Microwell chip	Bladder	Use of carbon metabolism to support biosynthesis	[53]
	Spheroids	Ovarian Gastric		[54,55]
**Autophagy employment as survival strategy**	3D culture basement membrane extracts reduced growth factor (Cultrex)	Breast	Use of opportunistic models of nutrient acquisition Elevated reliance on oxidative stress protection mechanisms	[56]
	Culture in low attachment plates	Prostate		[57]
**Use of macropinocytosis as a nutrient acquisition strategy**	Spheroids	Womb Epidermoid carcinoma	Use of opportunistic models of nutrient acquisition	[58]
**Use of entosis as a nutrient acquisition strategy**	Organoid	Ovary Colon	Use of opportunistic models of nutrient acquisition Increased demand of electron acceptors	[44,59]
	Spheroids	Ovary Colon Breast Liver		[45,46,47,48]
	3D bioprinting	Colon Breast Lung Pancreatic		[49,50,51]
**Alterations in the metabolic profile as a response to the hypoxia-induced redox stress**	3D silica scaffolds (CellBed^®^)	Prostate Bladder	Increased demand of electron acceptors Elevated reliance on oxidative stress protection mechanisms	[42]
	Organoids	Ovarian Colon		[44,54,60]
	Spheroids	Ovarian		[45,61]
	3D bioprinting	Lung Pancreatic Colon		[48,49]
**Reutilization of nitrogen from urea cycle**	Spheroids	Pancreatic	Increased demand for nitrogen	[62]
	Organoids	Breast		[63]
**Collagen composition of extracellular matrix increases cancer cell invasiveness**	3D collagen microchannels	Pancreatic	Heterogeneity of metabolic adaptations Metabolic interactions with the tumour microenvironment	[64]
	Spheroids	Breast		[65]
**Nutritional support by other cell populations**	Microfluidic device and scaffolds 3D scaffolds	Pancreatic Prostate	Heterogeneity of metabolic adaptations Metabolic interactions with the tumour microenvironment	[39,41]

## Data Availability

Not applicable.

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
