# Peer review of "Reflections on the Biology of Cell Culture Models: Living on the Edge of Oxidative Metabolism in Cancer Cells"

_ijms, 2023, doi:10.3390/ijms24032717_

Round 1
Reviewer 1 Report
I thank the authors for this interesting review.
I just suggest that the authors add table 1 earlier in the manuscript and refer to it more often in the text.
Author Response
We thank Reviewer 1 for this comment. We are very grateful for your efforts in reviewing our manuscript. In the new version of our manuscript, we have referred to Table 1 more often.
Reviewer 2 Report
In this review article, Moran-Alvarez et al. summarized the limitations of 2D culture models and discussed the utility of several 3D culture techniques to resolve those limitations. The topic of the review is of interest for the readership of this Journal, since the studies of cancer metabolism based on 3D culture is currently a hot topic in cancer research. The review is well-written. However, I have a few suggestions for the authors before it is published.
Organoids is the best option to develop 3D microenvironment in vitro. More information and discussion about this special 3D culture technique is encouraged.
As the authors stated that 3D culture has become an irreplaceable tool to improve cancer research in vitro. However, the issues and challenges in 3D culture techniques should be addressed as well.
Some typing errors in text should be corrected. For example, Line 288, “ADP+/NADPH”; Line 453, “500 mM in diameter”; Line 467, “effects.8. Conclusions‘’.
Author Response
In this review article, Moran-Alvarez et al. summarized the limitations of 2D culture models and discussed the utility of several 3D culture techniques to resolve those limitations. The topic of the review is of interest for the readership of this Journal, since the studies of cancer metabolism based on 3D culture is currently a hot topic in cancer research. The review is well-written. However, I have a few suggestions for the authors before it is published.
We thank the Reviewer for his/her important input.
Organoids is the best option to develop 3D microenvironment in vitro. More information and discussion about this special 3D culture technique is encouraged.
Following this suggestion, we have included a new paragraph summarizing the most used techniques and requirements to develop patient-derived organoids. The paragraph has been included right after the point where we state which are the most suitable technique(s) to reproduce tumour microenvironment (Section 8. Conclusions)
As the authors stated that 3D culture has become an irreplaceable tool to improve cancer research in vitro. However, the issues and challenges in 3D culture techniques should be addressed as well.
We understand the lack pointed out by the reviewer. Therefore, we have amplified section 7, discussing more specifically how 3D cultures would impact cancer metabolism research and further detailed some of their disadvantages and current technical limitations.
Some typing errors in text should be corrected. For example, Line 288, “ADP+/NADPH”; Line 453, “500 mM in diameter”; Line 467, “effects.8. Conclusions‘
Typing mistakes indicated on lines 288, 453, and 467 have been corrected.
Reviewer 3 Report
Paper reviews very important and emerging issues – the need of 3D cell culture system are more reliable to physiological condition. In general it is very well written, planned and finally covers very important issue – 3D cell culture and cancer cell metabolism. Nevertheless some comments remains after reading this manuscript and it will be valuable for authors to consider them.
One general remark is that the present work is very good in term of cell biology and metabolism but lacks biophysical and physical chemistry background which should be presented in such a review to provide a proper depiction of the topic.
Major:
1. Page 2 Line 62. Please reconsider if statement that maximal oxygen concentration within inner tissue is 5%. Authors refers only one work here what might be misleading. What is more even in this work authors provide higher concentration that 5% in inner tissue. While oxygen concentration in bone marrow is quiet low around 10 mmHg (10.1038/nature13034) it could be relatively higher in another tissues up to 72mmHg – what corresponds to around 10% in kidney (DOI:10.1213/00000539-199808000-00045, doi.org/10.1111/j.1582-4934.2011.01258.x). I agree with authors concept but it should be more detailed here and based on deeper literature lookup. It could be also valuable to address technical aspects of oxygen measurement is tissues and discuss how and it is possible to apply them in 3D cultures to obtain the same conditions.
2. While writing about impact of 3D culture system authors should refer to potential accumulation of metabolites in hydrogels/polymers used for culture (doi.org/10.1021/acsomega.1c03719, ofc thats jsut an example to show where to lookup. There are a lot of papers about accumulation/adsorption of molecules on biopolymers – please try to address this issue. It is very important. Also considering major point 3. In fact those two topic are interconnected but to some extent competitive. )
3. While considering significance of 3D cell-culture its seems that addressing physical properties of this system is missing. On one hand cytostatic drug treatment affects cancerous cells mechanical properies (doi.org/10.1039/D0NR06464E , doi.org/10.1038/s41598-022-23540-y doi.org/10.3390/ijms21228786) while on the other mechanical properties of cell surrounding could induce strong resistance against anticancer drugs (ultimately studied for breast cancer – but not only: doi.org/10.1016/j.biomaterials.2019.02.018, doi.org/10.1016/j.bbadis.2019.165625, 10.1038/s41586-022-05394-6)). I strongly recommend addressing paragraph for this topic. Introduce basic parameters i.e. elastic modulus, shear modulus, rheology, viscosity and techniques i.e. atomic force microscopy, elastography (10.1038/s41467-022-30995-0), rheology, traction force microscopy y e.t.c. And finaly several changes in physical properties of cells and tissues are associated with cancer progression (10.1038/s41586-022-05394-6)) and thus should be reflected while projecting 3D or 2Dhydrogel-based culture system for studying cellular metabolism.
4. I really appreciate the table summarizing works cited by authors in review. It will be very useful for readers looking for information about 3D cell culture systems for particular cancer types.
5. Section 7 “7. 3D models to improve cancer cell metabolism research” which should be in principle the key component of paper must be expanded. Authors should perform deeper literature lookup for this purpose.
Minor:
6. Page 1, 37 glutamine is not sugar. Please correct it.
7. Page 1, 41-16 Please precise what particular diagnostic techniques are used for particular molecules.
8. Page 3, 107 are authors sure that the term “epigenetic mutations” is correct?
9. While addressing paper (doi: 10.1038/ncomms16031) in line 127 at page 3 precise if this phenomena was observed in vitro or in vivo? Did atuhors report it only in context of serum albumin? How about collagen I and IV?\
10. Authors refers to very important issue – impact of cell culture medium on cellular metabolism (page 3, 149) (doi.org/10.1038/s41551-021-00775-0). It could be valuable to address this particular issue a little bit deeply. For istance although glutamine is non-essential amino acid in vivo, majority in vitro culture media relies strongly on glutamine supplementation ( i.e. 10.1021/bp049827d – just multiple studies are done in this subject so far). Since glutamine is for instance crucial for proste cancer development (glutamine addition) it will be valuable to discuss a little bit this issue (doi.org/10.1007/978-3-030-65768-0_2). On the other hand use of FBS is also introducing artifact to cell culture singe – apart from cows FBS is ultimately non-natural compound.
11. It could be also valuable to consider how particular types of 3D system culture affects pH (try to add table with references for instance) which is crucial for cellular metabolism.
12. Considering impact of forces in 3D culture – try to look in the literature for (potential) link between HIF(s) and mechanotransduction pathway. It might be valuable to address this topic.
13. Page 10 453 – “500 mM in diameter”, please correct it. Moles are rather not units of length. “imaging techniques such as microscopy and flow cytometry require employing expensive devices or destroying the extracellular structure before using fluorescent probes to understand the metabolic and redox landscape they develop” – this statement is quiet controversial. While use of flow cytometry in context of ECM-based hydrogel is very challenging, fluorescent microscopy is successfully applied here so far.
14. “Although the development of plastic structures that mimic tumour, stroma, and vasculature is still technically complex and expensive, they have successfully represented how breast cancer promotes immune suppression and recreate oxygenation gradient inducing VEGF secretion” – if you use PDMS for instance it is not so expensive.
Author Response
Response to Reviewer #3:
Paper reviews very important and emerging issues – the need of 3D cell culture system are more reliable to physiological condition. In general it is very well written, planned and finally covers very important issue – 3D cell culture and cancer cell metabolism. Nevertheless some comments remains after reading this manuscript and it will be valuable for authors to consider them.
One general remark is that the present work is very good in term of cell biology and metabolism but lacks biophysical and physical chemistry background which should be presented in such a review to provide a proper depiction of the topic.
We thank the Reviewer for his/her helpful comments, and we do completely agree with the limitations of the first version of our manuscript.
Major:
- Page 2 Line 62. Please reconsider if statement that maximal oxygen concentration within inner tissue is 5%. Authors refers only one work here what might be misleading. What is more even in this work authors provide higher concentration that 5% in inner tissue. While oxygen concentration in bone marrow is quiet low around 10 mmHg (10.1038/nature13034) it could be relatively higher in another tissues up to 72mmHg – what corresponds to around 10% in kidney (DOI:10.1213/00000539-199808000-00045, org/10.1111/j.1582-4934.2011.01258.x). I agree with authors concept but it should be more detailed here and based on deeper literature lookup. It could be also valuable to address technical aspects of oxygen measurement is tissues and discuss how and it is possible to apply them in 3D cultures to obtain the same conditions.
We agree that this issue might be controversial; therefore, the statement included on page 2, line 62 has been rephrased. Oxygen measurement strategies and the existing limitations in 3D culture have been added as part of the suggested amplification in section 7 (major suggestion 5)
- While writing about impact of 3D culture system authors should refer to potential accumulation of metabolites in hydrogels/polymers used for culture (doi.org/10.1021/acsomega.1c03719, ofc thats jsut an example to show where to lookup. There are a lot of papers about accumulation/adsorption of molecules on biopolymers – please try to address this issue. It is very important. Also considering major point 3. In fact those two topic are interconnected but to some extent competitive. )
Similarly to the previous issue, the potential accumulation of metabolites in hydrogels and polymers employed in 3D culture devices have been discussed in the suggested amplification of section 7.
- While considering significance of 3D cell-culture its seems that addressing physical properties of this system is missing. On one hand cytostatic drug treatment affects cancerous cells mechanical properies (doi.org/10.1039/D0NR06464E, doi.org/10.1038/s41598-022-23540-y doi.org/10.3390/ijms21228786) while on the other mechanical properties of cell surrounding could induce strong resistance against anticancer drugs (ultimately studied for breast cancer – but not only: doi.org/10.1016/j.biomaterials.2019.02.018, doi.org/10.1016/j.bbadis.2019.165625, 10.1038/s41586-022-05394-6)). I strongly recommend addressing paragraph for this topic. Introduce basic parameters i.e. elastic modulus, shear modulus, rheology, viscosity and techniques i.e. atomic force microscopy, elastography (10.1038/s41467-022-30995-0), rheology, traction force microscopy y e.t.c. And finaly several changes in physical properties of cells and tissues are associated with cancer progression (10.1038/s41586-022-05394-6)) and thus should be reflected while projecting 3D or 2Dhydrogel-based culture system for studying cellular metabolism.
As it has been suggested by the reviewer, a brief summary of the physical properties of 3D culture subtracts, and the techniques available to image them and measure those properties have been summarized at the end of section 7.
- I really appreciate the table summarizing works cited by authors in review. It will be very useful for readers looking for information about 3D cell culture systems for particular cancer types.
We appreciate this grateful comment considering this table a valuable resource for potential readers. We believe that the added literature listed might also answer minor suggestions 10 and 11).
- Section 7 “7. 3D models to improve cancer cell metabolism research” which should be in principle the key component of paper must be expanded. Authors should perform deeper literature lookup for this purpose.
As stated above, section 7 has been amplified, including relevant information that has been suggested on major suggestions 1, 2, and 3 and minor suggestion 12.
Minor:
- Page 1, 37 glutamine is not sugar. Please correct it.
We are sorry for this mistake. This statement has been rephrased.
- Page 1, 41-16 Please precise what particular diagnostic techniques are used for particular molecules.
As indicated, this specific information has been included in the paragraph.
- Page 3, 107 are authors sure that the term “epigenetic mutations” is correct?
This statement has been reformulated accordingly.
- While addressing paper (doi: 10.1038/ncomms16031) in line 127 at page 3 precise if this phenomena was observed in vitro or in vivo? Did atuhors report it only in context of serum albumin? How about collagen I and IV?\
This statement has been rephrased.
- Authors refers to very important issue – impact of cell culture medium on cellular metabolism (page 3, 149) (doi.org/10.1038/s41551-021-00775-0). It could be valuable to address this particular issue a little bit deeply. For instance although glutamine is non-essential amino acid in vivo, majority in vitroculture media relies strongly on glutamine supplementation ( i.e. 10.1021/bp049827d – just multiple studies are done in this subject so far). Since glutamine is for instance crucial for prostate cancer development (glutamine addition) it will be valuable to discuss a little bit this issue (doi.org/10.1007/978-3-030-65768-0_2). On the other hand use of FBS is also introducing artifact to cell culture singe – apart from cows FBS is ultimately non-natural compound.
- It could be also valuable to consider how particular types of 3D system culture affects pH (try to add table with references for instance) which is crucial for cellular metabolism.
(Response for points 10 and 11) We recognize the relevance of the shift towards particular nutrient sources, such as the use of carbohydrates as building blocks (named Warburg effect) or the increase in glutamine addiction in many cancer-cultured cells (section 1, lines 107-128, 139, including in the context of prostate cancer, among others, references 38-42). Both aspects greatly differ between 2D and 3D cultures, in both tumor cells and tumor microenvironment, where they cause changes in the pH, extracellular matrix stiffness, and cancer cell migration properties (References 43-45 and Table 1 and figure 2B). Even though we absolutely assume that both pH and certain nutrients addition, for instance, glutamine, are hallmarks of cancer cell metabolism and results about both parameters differ between 2D and 3D, we still consider it necessary to keep the balance between the content we report about those and other metabolic alterations. Nevertheless, it is important to insist that, considering the particular interest of those points, it was the reason why we have highlighted their implications in both, figure 2 and table 1.
- Considering impact of forces in 3D culture – try to look in the literature for (potential) link between HIF(s) and mechanotransduction pathway. It might be valuable to address this topic.
Reference to HIF(s) mechanotransduction pathway was included in the discussion about reproducing mechanical stimuli in vitro in section 7.
- Page 10 453 – “500 mM in diameter”, please correct it. Moles are rather not units of length. “imaging techniques such as microscopy and flow cytometry require employing expensive devices or destroying the extracellular structure before using fluorescent probes to understand the metabolic and redox landscape they develop” – this statement is quiet controversial. While use of flow cytometry in context of ECM-based hydrogel is very challenging, fluorescent microscopy is successfully applied here so far.
This statement has been reformulated.
- “Although the development of plastic structures that mimic tumour, stroma, and vasculature is still technically complex and expensive, they have successfully represented how breast cancer promotes immune suppression and recreate oxygenation gradient inducing VEGF secretion” – if you use PDMS for instance it is not so expensive.
This statement has been rephrased.
Round 2
Reviewer 2 Report
The revised manuscript is suitable for publication.
Reviewer 3 Report
The authors carefully addressed the issues underlined in my first-stage revision.
It is very valuable that authors referred to the most challenging topics while considering 3D culture methods in the context of cancer research i.e. accumulation of drugs within polymers used for cell culture and challenges in the release of cells from 3D cell culture (it is extremely important while thinking about the application of flow cytometry of cell sorting )
Taking into account all corrections done by authors I consider this work as very important to the field. It should be very useful for researchers interested in in vitro model of cancer in the context of its metabolism. Authors should be ultimately endorsed for high-quality figures and useful table.